# Upper Gastrointestinal Cancer and Liver Cirrhosis

**DOI:** 10.3390/cancers14092269

**Published:** 2022-05-02

**Authors:** Kuo-Shyang Jeng, Chiung-Fang Chang, I-Shyan Sheen, Chi-Juei Jeng, Chih-Hsuan Wang

**Affiliations:** 1Division of General Surgery, Far Eastern Memorial Hospital, New Taipei City 22060, Taiwan; cfchang.gina@gmail.com (C.-F.C.); joyce.walawala@gmail.com (C.-H.W.); 2Department of Medical Research, Far Eastern Memorial Hospital, New Taipei City 22060, Taiwan; 3Department of Hepato Gastroenterology, Chang-Gung Memorial Hospital, Linkou Medical Center, Taoyuan City 33305, Taiwan; happy95kevin@gmail.com; 4Postgraduate Institute of Medicine, National Taiwan University, Taipei 10617, Taiwan; b91401102@ntu.edu.tw

**Keywords:** upper gastrointestinal cancer, liver cirrhosis, gut microbiota, alcohol and tobacco, zinc

## Abstract

**Simple Summary:**

There is a higher incidence rate of upper gastrointestinal cancer in those with liver cirrhosis. The contributing factors include gastric ulcers, congestive gastropathy, zinc deficiency, alcohol drinking, tobacco use and gut microbiota. Most of the de novo malignancies that develop after liver transplantation for cirrhotic patients are upper gastrointestinal cancers. The surgical risk of upper gastrointestinal cancers in cirrhotic patients with advanced liver cirrhosis is higher.

**Abstract:**

The extended scope of upper gastrointestinal cancer can include esophageal cancer, gastric cancer and pancreatic cancer. A higher incidence rate of gastric cancer and esophageal cancer in patients with liver cirrhosis has been reported. It is attributable to four possible causes which exist in cirrhotic patients, including a higher prevalence of gastric ulcers and congestive gastropathy, zinc deficiency, alcohol drinking and tobacco use and coexisting gut microbiota. Helicobacter pylori infection enhances the development of gastric cancer. In addition, Helicobacter pylori, Porphyromonas gingivalis and Aggregatibacter actinomycetemcomitans also contribute to the development of pancreatic cancer in cirrhotic patients. Cirrhotic patients (especially those with alcoholic liver cirrhosis) who undergo liver transplantation have a higher overall risk of developing de novo malignancies. Most de novo malignancies are upper gastrointestinal malignancies. The prognosis is usually poor. Considering the surgical risk of upper gastrointestinal cancer among those with liver cirrhosis, a radical gastrectomy with D1 or D2 lymph node dissection can be undertaken in Child class A patients. D1 lymph node dissection can be performed in Child class B patients. Endoscopic submucosal dissection for gastric cancer or esophageal cancer can be undertaken safely in selected cirrhotic patients. In Child class C patients, a radical gastrectomy is potentially fatal. Pancreatic radical surgery should be avoided in those with liver cirrhosis with Child class B or a MELD score over 15. The current review focuses on the recent reports on some factors in liver cirrhosis that contribute to the development of upper gastrointestinal cancer. Quitting alcohol drinking and tobacco use is important. How to decrease the risk of the development of gastrointestinal cancer in those with liver cirrhosis remains a challenging problem.

## 1. Introduction

The extended scope of upper gastrointestinal tract includes the esophagus, stomach (including cardia), duodenum, small intestine, pancreas, bile ducts and liver. In this review, cancer of liver and bile duct will be excluded. A higher incidence of some gastrointestinal cancers in patients with liver cirrhosis has been reported [1,2]. This review will explore four major aspects of this issue. First, the factors associated with liver cirrhosis can contribute to the development of upper gastrointestinal cancer. Second, it will explore the role of some gut microbiota involved in the gut–liver axis in cirrhotic patients in the development of upper gastrointestinal cancer. Third, it will discuss de novo gastrointestinal cancer that develops after liver transplantation performed on those with liver cirrhosis, especially alcoholic liver cirrhosis. Finally, the challenges of surgical treatment of upper gastrointestinal cancer in those with liver cirrhosis will be considered.

## 2. Factors Associated with Liver Cirrhosis Contributing to the Development of Upper Gastrointestinal Cancer

The prevalence of upper gastrointestinal cancer in those with liver cirrhosis is higher than that in those without liver cirrhosis [1,2]. A significant 2.6-fold (*p* < 0.01) prevalence of gastric cancer in cirrhotic patients has been reported [2]. According to Kalaitzakis et al.’s report, compared with the general population, cirrhotic patients have an increased risk of esophageal cancer (odds ratio 8.3, 95% CI 1.7–24.2) and pancreatic cancer (odds ratio 5.1, 95% CI 1.4–13.2) [3]. The prevalence of pancreatic cancer is also significantly elevated in those with primary biliary cirrhosis [4,5].

The high incidence rate of gastric cancer and esophageal cancer in patients with liver cirrhosis can be attributed to four possible causes (Table 1). The first is a higher prevalence of gastric erosions, gastric ulcers and congestive gastropathy in cirrhotic patients [2,6,7,8,9]. The second is zinc deficiency in cirrhotic patients [10,11,12,13]. The third is alcohol drinking and tobacco use [14,15,16,17,18,19,20,21,22,23,24,25,26,27,28,29,30,31,32,33,34,35,36,37,38,39,40,41,42,43,44,45,46,47,48,49,50,51,52]. Finally, the coexisting gut microbiota in patients with cirrhosis can contribute to the development of upper gastrointestinal cancer.

### 2.1. Higher Prevalence of Gastric Erosions, Gastric Ulcers and Congestive Gastropathy in Cirrhotic Patients

Gastric ulcer increases the potential for gastric cancer development [5,6]. Voulgaris et al. reported that the prevalence of peptic ulcers among those with liver cirrhosis is as high as 19% [6]. Gastric ulcers are more common than duodenal ulcers [6]. Zullo et al. also suggested a higher frequency of gastric erosions and gastric ulcers in those with liver cirrhosis [2,53]. The coexisting peptic ulcer disease does not correlate with the severity or the etiology of cirrhosis [6]. In addition, portal hypertensive gastropathy exists in the majority of patients with liver cirrhosis [6]. In those with severe portal hypertensive gastropathy, gastric ulcer prevalence is higher [6]. Congestive gastropathy due to liver cirrhosis can significantly facilitate the proliferation of epithelial cells in gastric mucosa [2,7,9].

Endoscopic injection sclerotherapy is a well-established treatment for esophageal varices often occurring in cirrhotic patients. There is no evidence of a direct correlation between sclerotherapy and esophageal cancer [8]. However, Ohta et al. suggested that in those with alcoholic cirrhosis with a risk of esophageal cancer, the chronic inflammation resulting from sclerotherapy can accelerate the malignant potential [8]. The progress of such patients should therefore be closely monitored with an endoscopy.

Gastric erosions, gastric ulcers, congestive coagulopathy and sclerotherapy of varices all contribute to gastric and esophageal cancers.

### 2.2. Possible Contribution of Zinc Deficiency in Cirrhotic Patients to Upper Gastrointestinal Cancer

Among the trace elements in the human body, zinc is the second most abundant trace element in the human body, after only iron [10]. Zinc is involved in more than 300 enzymes [10]. Zinc plays a pivotal role in the growth, differentiation and metabolism of cells [10]. The enzymes for both collagen production and collagen destruction involved in the fibrosis process are affected by zinc and zinc plays a role in the anti-fibrotic pathway [10]. Zinc also has anti- inflammatory and antioxidant characteristics which can affect hepatic stellate cells indirectly [9,10,11,12,13,54]. According to Kodama et al.’s suggestion, serum zinc <60 μg/dL is defined as a definite deficiency and 60–80 μg/dL is defined as a marginal deficiency [55]. Zinc deficiency in patients with liver cirrhosis is multifactorial [2,13,56,57]. Zinc is bound to serum albumin, alpha 2-macroglobulin and acids [11]. The absorption of zinc is affected by the albumin level [11]. After the progression of liver cirrhosis, the albumin level decreases and the zinc deficiency is aggravated [11]. The increase in zinc excretion after the use of diuretics to treat ascites also causes a zinc deficiency [11]. Other factors affecting the intestinal mucosa of zinc absorption include endotoxins on gut blood flow and the cytokines, especially interleukin-6 [11]. Changes in carbohydrate–lipid metabolism and protein-calorie malnutrition occurring in more than 60% of patients with severe alcoholic cirrhosis cause micronutrient malnutrition including trace elements [57,58,59]. In cirrhotic patients, zinc deficiency induces a nitrogen metabolic disorder and can affect growth disorders, oxidative stress, cognitive disorders and immune dysfunction [10,55]. Zinc replenishment can be beneficial to antioxidant and inflammatory pathways and can mitigate the progression of cirrhosis [60,61].

Zinc deficiency enhances both the epithelial carcinogenesis and tumor progression of esophageal cancer via microRNA expression [62,63]. From a rodent experiment, Fong et al. found that a zinc deficiency increases the risk of upper aerodigestive tract cancer [64]. The mechanism is more than the cyclooxygenase (COX) 2 pathway [64].

Some authors have suggested that the upregulation of both zinc influx transporters (ZIPs) and zinc transporters occurs in many gastrointestinal cancers, though some controversy still exists [65]. Kumar et al. found the upregulation of ZIP7 in esophageal squamous cancer cells [66]. The dysregulation of zinc transporters exists in gastrointestinal cancers including pancreatic cancer [64]. The upregulation of zinc transporters can enhance the migration of pancreatic cancer cells and can lead to a poor prognosis [67]. This is especially true for ZIP4, which could be used as a diagnostic and prognostic marker [68,69]. Li et al. found that the overexpression of ZIP4 mRNA is present in most (16 of 17) surgical specimens of pancreatic adenocarcinoma [67]. Xu et al. also reported that the upregulation of ZIP4 occurred in 23 of their pancreatic cancer samples [70]. Tumors grew more rapidly in the ZIP4-expressing xenografts in mice [71].

Taccioli et al.’s rat experiments showed that a zinc supplement can reverse the overexpression of S100A8, a proinflammatory mediator in esophageal preneoplasia [72]. Fong et al. also suggested that zinc supplementation can prevent upper aerodigestive tract cancer [64]. Jin et al. [73] reported that the knockdown of ZIP5 significantly inhibited the cell progression of human esophageal squamous cell carcinoma [73]. Choi et al. found that a zinc supplement mitigated the cell proliferation of esophageal squamous cancer cells via Orai1-mediated store-operated calcium entry (SOCE) and also the subsequent intracellular Ca^2+^ oscillations [74].

### 2.3. Alcohol Drinking and Tobacco Use in Alcoholic Liver Cirrhotic Patients Can Accelerate the Malignant Potentials of the Upper Gastrointestinal Tract

#### 2.3.1. Alcoholic Drinking

Alcohol drinking is the main contributing factor to alcoholic liver cirrhosis. Higher levels of alcohol intake (from 45 gm of alcohol per day) increase the risk of gastric cancer, including both cardia and non-cardia gastric cancers [33]. Even about 100 gm of alcohol per week or less could currently be the limit of low-risk use [33]. The amount and duration of alcohol drinking increases the risk of the cancer. The World Cancer Research Fund (WCRF) and the American Institute of Cancer Research (AICR) have suggested that from many Asian studies, there is a positive association between alcohol drinking and risk of gastric cancer [33]. Alcohol also causally increases the risk of oesophageal squamous cell cancer and pancreatic cancer [14,75].

Alcohol undergoes chemical coupling to membrane phospholipids and disrupts the organization of tight junctions, leading to the nuclear translocation of β-catenin and zonula occludes-associated nucleic acid binding proteins (ZONAB) to manipulate the genes involving the proliferation, invasion and metastasis of cancer [34,35,36]. Alcohol enhances reactive oxygen species (ROS) generation and disturbs the function of scavenger systems, facilitating oxidative stress and resulting in the instability of genes [34,35,36]. In addition, alcohol inhibits antioxidant activity and cytoprotective enzymes but enhances the activity of CYP2E1 to induce the metabolic activation of chemical carcinogens [33]. Ethanol can be metabolized by alcohol dehydrogenases (ADH), cytochrome P450 2E1 (CYP2E1) or catalase to acetaldehyde. Acetaldehyde then can be oxidized to acetate by aldehyde dehydrogenase (ALDH). According to the International Agency for Research on Cancer (IARC), acetaldehyde has been classified as a Group 1 carcinogen to humans [37]. Acetaldehyde is also a recognized carcinogen in experimental animal models [15]. Alcohol enhances the susceptibility of various organs to chemical carcinogens to uptake the various metabolites to alter the composition of enteric microbes to elevate the aldehyde level. It activates procarcinogens to facilitate the changes in the microsomal enzyme in the metabolism and distribution of carcinogens. It can weaken the repair system of carcinogen-induced DNA alkylations and can shorten telomere length [34,35,36]. In the upper gastrointestinal tract, the produced acetaldehyde and free radicals via cytochrome P450 2E1 and via aldehyde can damage the mucosal tissue and trigger the repeated cellular regeneration. The acetaldehyde produced by local aldehyde dehydrogenase (ALDH) can also enhance cancer development [16]. Polymorphisms in the alcohol metabolizing enzyme especially aldehyde dehydrogenase-2 (ALDH2) can affect salivary acetaldehyde concentrations after alcohol consumption [17,18,27]. Although acetaldehyde is mainly produced in the liver, acetaldehyde formation begins in the mouth and continues along the digestive tract [14]. Intragastric acetaldehyde level is locally regulated by both the gastric mucosal ADH and ALDH2, and the microbes colonizing the stomach and saliva [14,26]. The microbiome plays a pivotal role in alcohol use-related gastrointestinal carcinogenesis [34,35,37]. Because Helicobacter pylori in both the gastric mucosa and oral bacteria can produce acetaldehyde, the gastric level of acetaldehyde can be affected by the gastric colonization of Helicobacter pylori [14].

There are mutations of the ALDH enzyme. Those with the genotype of ALDH2-2 allele or ADH1C*1/1, (genetic predisposition for alcohol-mediated cancer,) pose a higher risk for developing esophageal cancer [23,33,38,39] The variant ALDH2*2 allele is a genetic risk of smoke and alcohol-induced esophageal cancer (including esophageal squamous cell carcinoma), gastric cancer and pancreatic cancer [23,24,25]. The mutated ALDH2–2 allele was an inactive form of ALDH. Koyanagi et al. found a significantly increased risk for esophageal cancer and gastric cancer from the direct effect of ALDH2 Lys allele [21]. Those with a mutated ALDH2–2 allele accumulated acetaldehyde had a higher risk of alcohol-related cancers than those with a wild-type allele after alcohol ingestion [21,23,38]. ALDH2–2 mutations are found commonly in (East) Asian people especially the ALDH2 polymorphism (rs 671) [20,21,22,23,38]. People with the genotype (ALDH2.2 allele) also present a genetic predisposition for smoke and alcohol-related cancers [23,24,25,26,39]. A genetic variant of aldehyde dehydrogenase 2 (ALDH2 rs671, Glu504Lys) can also contribute to cancer development in alcohol drinkers [20]. Some epidemiological studies have shown that ALDH2 can trigger both carcinogenesis and the progression or metastasis [20].

#### 2.3.2. Tobacco Smoking

Xiong et al. suggested that smoking triggers a fibrogenic effect on the liver [76]. The apoptosis, oxidative stress and hypoxia induced by smoking can increase hepatocellular damage [77]. Nicotine, the main constituent of cigarette smoke, can stimulate hepatic stellate cells and up-regulate the fibrogenic markers, TGF-β, and collagen [78]. Smoking enhances the production of pro-inflammatory cytokines in the peripheral blood which can induce the progression of chronic hepatitis B [79,80,81]. Among HBeAg-negative cases, Xiong et al. found that patients who smoked had significantly higher DNA loads of hepatitis B virus than those who did not smoke, suggesting that the smoking-induced HBV DNA burden upon HBeAg-negative patients can contribute to advanced liver fibrosis or cirrhosis [76].

Those who smoke have about a 1.62-fold higher risk of gastric cancer (95% CI: 1.50–1.74) than nonsmokers in Chinese people [31]. Some meta-analyses have shown that the risks of esophageal and gastric cardia adenocarcinoma for smokers are similar [28]. Either tobacco or alcohol use resulted in a 20–30% risk for esophageal squamous cell carcinoma (ESCC) compared with non-use. Both tobacco and alcohol use resulted in as high as a 3-fold risk of ESCC [17]. Acetaldehyde also exists in mainstream tobacco smoke [30]. Long-term smoking can increase the levels of acetaldehyde in saliva to modify oral flora after alcohol ingestion [29]. Some Helicobacter. pylori strains retain substantial cytosolic ADH activity and can produce bulky amounts of acetaldehyde after incubation with ethanol [26,82]. Smoking with ethanol consumption increases acetaldehyde 7-fold compared with ingestion alone, suggesting the synergistic risk of alcohol drinking and tobacco smoking for cancer development in the upper gastrointestinal tract [17,31,32].

Alcohol drinking is one important risk factor for pancreatic cancer, with a population attributable risk of 3% [83,84]. The combined use of tobacco smoking and alcohol increases the cumulative risk of pancreatic cancer [83,85,86].

Alcohol drinking and tobacco use contributing to liver cirrhosis also enhances the development of esophageal cancer, gastric cancer and pancreatic cancer. Preventing or discontinuing the use of alcohol and tobacco is important in decreasing the cancer risks.

### 2.4. The Oral–Gut–Liver Axis-Involved Microbiota in Cirrhotic Patients Contributes to the Development of Upper Gastrointestinal Cancer

The normal gut microbiota play a pivotal role in the nutrient metabolism of the host and in maintaining the structural integrity of the mucosa barrier and immunomodulation of the gut [87,88]. It has been hypothesized that gut dysbiosis could trigger the development of some types of cancer via systemic mechanisms, including metabolic changes to affect precancerous cells and immune cells [88,89].

The portal vein links the intestinal microbiome and the liver via bidirectional interactions. Dysbiosis of the intestinal microbiome can affect the progression of liver diseases to liver cirrhosis and its complications [89,90].

The gut microbiota changes via the impairment of the gut–liver axis in those with cirrhosis [91]. Alterations in microbiota result from the disruption of some factors in liver cirrhosis including reduced (i) small bowel motility and transit time, especially in the ascitic state, as one main contributor to dysbiosis; (ii) bile acid abnormalities, consisting of decreased primary bile acid and increased secondary bile acid within the gut; and (iii) impaired intestinal immunity [92,93,94,95,96,97,98]. Feng et al.’s analysis suggested that there is a significantly high prevalence of H. pylori infection in patients with cirrhosis [99]. H. pylori infection occurs significantly more frequently among those with liver cirrhosis (hepatitis C virus infection or hepatitis B virus infection) than those with alcoholic cirrhosis or primary biliary cirrhosis [99,100].

Three mechanisms for the role of oral microbiota in the pathogenesis of cancer have been suggested [100,101,102]. The first is that bacterial stimulation induces chronic inflammation. Inflammatory mediators can facilitate cell proliferation, mutagenesis, angiogenesis and oncogene activation [100,101,102]. The second mechanism is that bacteria affect cell proliferation via the activation of NF-κB and the inhibition of cellular apoptosis [100,101,102]. In the third mechanism, some substances produced by bacteria can have a carcinogenic effect [100,101,102]. Michaud et al. examined the relationship between antibodies to 25 oral bacteria and pancreatic cancer risk in a prospective cohort study [103].

In those with severe liver cirrhosis, H. pylori infection activates kupffer cells and hydrogen peroxide to enhance TGF-β1 to trigger pro-inflammatory signaling pathways in hepatic stellate cells (HSC) to release cytokines [104,105,106,107].

The studies of liver samples from cirrhotic patients with HCV-infection showed that the cagA gene was more prevalent in advanced cirrhosis (28.2%) compared to early fibrosis (5.9%) [108]. The hepatocytes can be altered by H. pylori infection, causing in collagen accumulation with liver fibrosis [109].

Elevated serum levels and liver tissue levels of FoxP3 and RORγt H.pylori-infected hepatitis B were found in cirrhosis patients, suggesting deteriorated liver damage [110]. Helicobacter pylori infection as the major risk factor for gastric cancer development is well known [111]. H. pylori-induce inflammation can damage the mucosa barrier to trigger gastric carcinogenesis after precipitating factors (such as tobacco smoking) or the genetic disposition of the host [111]. Polymorphisms and epigenetic changes of the host gene coding (for interleukins (IL1β, IL8), transcription factors (CDX2, RUNX3) and DNA repair enzymes) and the genetic variance of bacterial proteins (CagA and VacA, etc.) increase the gastric cancer risk [111].

Zaidi et al. reported that a large amount of Escherichia coli existing in esophageal tissues can enhance the expression of the toll-like receptor signaling pathway to trigger Barrett’s esophagus and esophageal carcinoma [112]. A significant amount of Campylobacter concisus in esophageal tissues can also affect Barrett’s esophagus with the overexpression of IL18 to induce carcinogenesis [113]. Yamamura et al. found that a large amount of Fusobacterium nucleatum affects the expression of chemokine CCL20 of the tumor and affects aggressive tumor behavior via the activation of chemokines in esophageal carcinomas [114]. This can lead to a shorter cancer-specific survival and overall survival in both esophageal squamous cell carcinoma (ESCC) and esophageal adenocarcinoma [114].

Some gut bacteria play a possible role in pancreatic cancer. [104] Nagao et al. emphasized the correlation between the periodontal diseases and hepatitis B- and C-related liver cirrhosis [115]. Salivary investigations showed an increase in Enterobacteriaceae and Enterococcacea in cirrhotic patients [116]. The mouth wash samples study demonstrated that a higher abundance of P. gingivalis and Aggregatibaoter actinomycetemcomitans (A. actinomycetemcomitans), associated with a decreased relative abundance of Fusobacterium and its genus Leptotrichia. All those findings may increase the risk of pancreatic cancer [117]. A high amount of the P. gingivalis fimbrillin A genotype exists in the saliva of those with liver cirrhosis [118,119,120]. Mohammed et al. suggested that the carriage of the periodontal pathogens P. gingivalis and A. actinomycetemcomitans increases the risk of pancreatic cancer [87]. They also reported that some oral bacteria, particularly P. gingivalis, with its elevated blood serum antibodies, pose a higher risk of both pancreatic cancer and liver cirrhosis [87]. The risk of pancreatic cancer increases in those with elevated levels of blood serum antibodies for select oral pathogens—namely, P. gingivalis as well as higher levels of genotype fimbrillin P. gingivalis—in the saliva of patients with liver cirrhosis [87,115,121,122,123,124,125,126]. P. gingivalis can disrupt the host immune system and affect some signaling pathways through cytokine and receptor degradation [117].

Some studies have found that those with high levels of antibodies against P. gingivalis ATTC 53978 had a higher risk of pancreatic cancer [103]. P. gingivalis and A. actinomycetemcomitans can activate Toll-like receptor signaling pathways. From animal models, Toll-like receptor activation is a pivotal promoter of pancreatic cancer [117,127]. Barton et al. found that a specific mutation in the cell cycle controller p53 occurring in those with pancreatic cancer can lead to the loss of arginine [128]. A recent study showed that the bacterial peptidyl arginine deaminase (PAD) enzyme affects this mutation in patients with pancreatic cancer. Porphyromonas gingivalis, Prevotella intermedia, Treponema denticola and Tannerella forsythia all hold this PAD enzyme. The activity of the PAD enzyme is associated with the modification of the Pro allele p53Arg72-Pro that can trigger the development of pancreatic cancer [129,130]. Another Fusobacteria genus (Alloprevotella) also plays a role in the risk of pancreatic cancer [117]. Some studies have found that the oral periodontal pathogens Fusobacterium nucleatum and Porphyromonas gingivalis play an important role in the development of pancreatic cancer [129,130,131].

It has been suggested that H. pylori infection is involved in the acute and chronic pancreatitis pathogenesis, autoimmune pancreatitis, diabetes mellitus and metabolic syndrome [104]. Rabelo-Gonçalves et al. suggested that H. pylori infection plays a potential role in the development of pancreatic cancer [104,131]. Ai. F et al. emphasized that H. pylori infection, especially with Cag A positive strains, is a risk factor for pancreatic carcinogenesis [131]. Some mechanisms of H. pylori enhancing pancreatic cancer have been proposed [132,133,134,135,136,137,138]. H. pylori gastritis can increase gastrin and somatostatin and enhance DNA synthesis [132,133,134,135]. Bacterial overgrowth can increase N-nitroso components and chronic inflammatory changes [136,137,138]. H. pylori chronic infection increases reactive oxygen species and proinflammatory cytokines and other inflammatory mediators. The enhanced cell proliferation and genomic DNA damage, with the inactivation of tumor-suppressor genes, can cause the malignant transformation of pancreatic cells [101]. Takayama et al. found after H. pylori infection, there are increased activities of activator protein-1, nuclear factor-kb, the serum level of IL-8 and increased serum response element of human pancreatic cancer cells [102]. This suggests that the development of pancreatic cancer could be similar to the gastric carcinogenesis. The environmental factors that enhance the development of pancreatic cancer and gastric cancer are similar, including smoking, alcohol consumption and dietary habits [100].

The gut microbiota in cirrhotic patients triggers the development and progression of esophageal cancer, gastric cancer and pancreatic cancer.

## 3. De Novo Upper Gastrointestinal Malignancies in Recipients with Liver Cirrhosis after Liver Transplantation

The patients who underwent liver transplantation had a higher overall risk of developing de novo malignancies than the general population [139,140,141,142,143,144,145,146,147]. The majority of these patients were male and had alcohol-related liver cirrhosis [139,140,141]. Prior to transplantation, some investigators reported that tobacco use was 52% to 83.3% in alcoholic cirrhotic patients [141]. Most of them developed upper gastrointestinal malignancies (esophagus, stomach) [139,140,141,142,143,144,145,146]. However, most of the de novo malignancies are diagnosed at an advanced stage (≥III).One-year survival (about 50%) and total survival (about 28.6%) are poor [139].

Immunosuppressive medications after transplantation also increase the risk of de novo malignancies. They can cause direct damage of the host DNA to impair the immune competence of the recipient [142,143]. Aside from immunosuppression and a history of alcoholic abuse and tobacco smoking, other identified risk factors for de novo malignancies include the patient’s age, primary sclerosing cholangitis and viral infections with oncogenic potential [140,141,142,143,144,145].

## 4. Is the Risk of Surgical Treatment of Upper Gastrointestinal Cancer in Patients with Advanced Liver Cirrhosis Higher?

Consideration of the surgical risk of upper gastrointestinal cancer among those with liver cirrhosis is important. Resection of upper gastrointestinal cancer in cirrhotic patients is usually associated with poor postoperative outcomes [148,149,150]. The severity of liver cirrhosis is the primary determinant of postoperative mortality [148,149,150,151].

Mortality is high in those with moderate to severe liver dysfunction. On multivariate analysis, cirrhosis was an independent predictor of in-hospital mortality and longer lengths of stay and high possibilities of long-term care facilities after discharge [148]. However, liver cirrhosis is not an absolute contraindication for surgery of upper gastrointestinal cancer. Gastrointestinal cancer operations can be performed safely in well-selected cirrhotic patients with mild liver dysfunction [148].

The model for end-stage liver disease (MELD) score can be applicable for an esophagectomy risk assessment for cirrhotic patients [150]. According to Valmasoni et al., those with a MELD score of nine or lower showed an outcome similar to that of the noncirrhotic patients [150]. Schizas et al. also emphasized that esophagectomies for esophageal carcinoma in Child–Turcotte–Pugh class (Child class) A cirrhotic patients have significantly lower 30-day mortality rates than the B patients [152]. Alshahrani et al. reported that in the case of early gastric cancer, for those with Child class A liver cirrhosis, a laparoscopic or laparoscopy-assisted distal gastrectomy can be as safe as an open distal gastrectomy with a similar long-term survival rate and immediate postoperative liver function [153]. According to Guo et al., a radical operation with a D1 or D2 lymph node dissection could be undertaken in Child class A gastric cancer patients [154]. A D1 lymph node dissection could be performed in Child class B patients [154]. However, in Child class C patients, a radical gastrectomy is very dangerous, even fatal [154]. Schwarz et al. stated that for pancreatic radical surgery in those with liver cirrhosis, Child–Pugh classes B and a MELD score value over 15 could be associated with a higher morbidity and mortality [155]. Radical surgery of the pancreas should be avoided [155].

Endoscopic submucosal dissection (ESD) of esophageal or gastric neoplastic lesions in patients with liver cirrhosis has been reported [149,150,151,156,157,158,159,160,161,162]. Repici et al. undertook an endoscopic submucosal dissection of gastric neoplastic lesions in those with liver cirrhosis [151]. Their successful rate of en bloc removal and the R0 resection were 88.2% and 89.7%, respectively, with complications of bleeding (13.1%) and perforation (1.6%) [151]. No procedure-related deaths were observed. Patients with advanced cirrhosis, with either INR >1.33 and/or a platelet count <105,000/mm^3^ should be regarded as having an increased risk of bleeding following ESD. ESD-related bleeding occurred more frequently in Child–Pugh class B/C patients as compared to those in class A (5/9 vs. 1/33; *p* < 0.001, Fisher’s exact test) [151]. They emphasized that all these complications can be successfully managed by endoscopy [151]. ESD for gastric neoplastic lesions in cirrhotics is effective and relatively safe [151]. A procedure-related complication is a bleeding, but it can be successfully controlled endoscopically [151].

ESD, initially developed for gastric cancer, is currently accepted for superficial cancer of the esophagus [156]. The most important advantage of ESD, compared with endoscopic mucosal resection (EMR), is that ESD can provide a higher en bloc resection rate and a precise histologic assessment, including in the case of large lesions [157]. ESD poses a higher risk of bleeding and perforation than endoscopic mucosal resection (EMR) [158,159,160,161,162].

However, ESD of esophageal cancer or gastric cancer for patients with cirrhosis still carries a higher risk of these adverse events because of the low platelet count, coagulopathy and portal hypertensive gastroenteropathy, including esophageal varices. The esophageal cancer patients to receive ESD should be selected.

## 5. Conclusions

Many different factors contribute to the development of upper gastrointestinal tract cancer (including esophagus, stomach and pancreas). Some factors exist in those with liver cirrhosis including gastric erosion, gastric ulcer, congestive gastropathy, zinc deficiency, alcohol drinking, tobacco use, infection of Helicobacter pylori, Porphyromonas gingivailis and Aggregatibacter actinomycetemcomitans. Preventing or minimizing these factors could avoid or mitigate the occurrence or progression of cancer. Quitting alcohol drinking and tobacco use could be important. Some factors are complex. How to decrease the risk of the development of gastrointestinal cancer in those with liver cirrhosis remains a challenging problem.

## Figures and Tables

**Table 1 cancers-14-02269-t001:** Factors associated with liver cirrhosis contribute to upper gastrointestinal cancer.

Factors	Possible Mechanism
Gastric erosion/ulcers, congestive gastropathy	chronic inflammation
Zinc deficiency	antifibrotic pathway antioxidant and inflammatory pathway cytokines (interleukin-6) endotoxins on gut blood flow zinc influx transports, zinc transporter
Alcohol consumption	tight junctions nuclear translocation of β catenin, ZONAB oxidative stress gene instability ADH ALDH ALDH2*-2 allele, ADH 1C*1/1
Tobacco use	ADH H. pylori
Gut microbiota	Gut–liver axis TGF-β pathway genes chemokines gastrin, somatostatin toll-like receptor signaling pathway PDA enzymes NF-κB activation cellular apoptosis

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
