# Peer review of "Upper Gastrointestinal Cancer and Liver Cirrhosis"

_cancers, 2022, doi:10.3390/cancers14092269_

Round 1

Reviewer 1 Report

The review is well written and an provides a mechanistic perspective that is often lacking in many cancer incidence reviews.

As a suggestion to the authors, in subsections throughout section 2 of the manuscript was the use of "increased risk" followed by the corresponding reference support. It would aid the reader greatly if an available and appropriate numerical values were given. One might consider adding the appropriate odds ratio, percentage increase, etcetera. Stating a numerical value often strengthens correlative arguments rather than use of the nondescriptive “increased risk” term.

(Line 126) Zinc deficiency in cirrhotic patients contributes to upper gastrointestinal cancer.

The sentence appears wanting. Is there additional sentence or summation that was omitted? If true please consider revising.

Minor: The reviewer noted two typos, line 87, anti- inflammatory and line 290 please use cause instead of causes.

Author Response

Dear reviewer:

Thank you very much for your comments. I reply as followings.

Comments:

As a suggestion to the authors, in subsections throughout section 2 of the manuscript was the use of “increased risk” followed by the corresponding reference support. It would aid the reader greatly if an available and appropriate numerical values were given. One might consider adding the appropriate odds ratio, percentage increase, etcetera. Stating a numerical value often strengthens correlative arguments rather than use of the nondescriptive “increased risk” term. (Line 126) Zinc drficiency in cirrhotic patients contributes to upper gastrointestinal cancer.

The sentence appears wanting. Is there additional sentence or summation that was omitted? If true please consider revising.

Reply:

Comments:

Minor: The reviewer noted two typos, line 87, anti- inflammatory and line 290, please use cause instead of causes.

Reply:

I corrected to anti-inflammatory.

I corrected it, using cause.

Reviewer 2 Report

This is a review regarding correlation of upper gastrointestinal cancers and liver cirrhosis (LC). This review describes the factors associated with LC and associated carcinogenesis, especially the role of gut microbiota involved in the gut-liver axis in LC, de novo gastrointestinal cancer developing after liver transplantation performed on those with LC, and surgical treatment of upper gastrointestinal cancer with LC. I consider this review has potential of publication; however, I have a few comments.

In introduction, why cancer of liver and bile duct are excluded? The reason should be documented.

In section 2 (Factors associated with liver cirrhosis contributing to the development of upper gastrointestinal cancer), the section of Pancreatic Cancer (2.3.3.) is not suitable for independent section. This document should be included in Alcoholic drinking section (2.3.1.).

The section 4 (Is the risk of surgical treatment of upper gastrointestinal cancer in patients with liver cirrhosis higher?), the description in this section is not correctly reflect subtitle. More focused description regarding the topic of subtitle should be clearly documented. For example, summary of previous studies compared surgical outcome between LC patient group and non-LC patient group in various organs of various surgical procedure would produce useful information for readers.

Author Response

Thank you very much for your comments. I reply as followings.

Comments:

In Introduction, why cancer of liver and bile duct are excluded? The reason should be documented.

Reply:

I had asked the Editor Office, whether the category of upper gastrointestinal cancers include or exclude liver via mail on 22 June 2021. The Editor answered that the liver cancers were excluded.

Comments:

In section 2, (Factors associated with liver cirrhosis contributing to the development of upper gastrointestinal cancer), the section of Pancreatic cancer (2.3.3) is not suitable for independent section. This document should be included in Alcoholic drinking section (2.3.1).

Reply:

 Comments:

The section 4 (Is the risk of surgical treatment of upper gastrointestinal cancer in patients with liver cirrhosis higher?), the description in this section is not correctly reflect subtitle. More focused description regarding the topic of subtitle should be clearly documented. For example, summary of previous studies compared surgical outcomes between LC patient group and non-LC patient group in various organs of various surgical procedure would produce useful information for readers.

 Reply:

  1. Is the risk of surgical treatment of upper gastrointestinal cancer in patients with liver cirrhosis higher?

I corrected the subtitle to “……….with advanced liver cirrhosis higher?”

In this subtitle, we mentioned the impact of severity of cirrhosis on the surgical outcomes, as followings.

Line 345-346: Gastrointestinal cancer operations can be performed safely in well-selected cirrhotic patients with mild liver dysfunction [148].Line 359-361: Schwarz et al stated that for pancreatic radical surgery in those with liver cirrhosis, Child-Pugh classes B and a MELD score value over 15 could be associated with higher morbidity and mortality [155]. Line 368-370: ESD related bleeding occurred more frequently in Child-Pugh class B/C patients as compared to those in class A (5/9 vs 1/33, p<0.001, Fisher's exact test). I will add this into revised manuscript.

Round 2

Reviewer 2 Report

Although the revised parts are difficult to understand due to the display style of changed track, I could confirm all the changes which the authors have made. The authors have correctly responded to the reviewer’s comments. I have no further comment.